# Use of allele-specific qPCR and PCR-RFLP analysis for rapid detection of the SARS-CoV-2 variants in Tunisia: A cheap flexible approach adapted for developing countries

Wajdi Ayadi[1]*, Fahmi Smaoui[2], Saba Gargouri[2,3], Sana Ferjani[4,5], Zaineb Hamzaoui[5], Awatef Taktak[1,2], Amel Chtourou[2,3], Houda Skouri-Gargouri[1], Azza Hadj Sassi[1], Mouna Ben Sassi[6,7], Sameh Trabelsi[6,7], Ali Gargouri[1], Ilhem Boutiba-Ben Boubaker[4,5], Héla Karray-Hakim[2,3], Raja Mokdad-Gargouri[1], Lamia Feki-Berrajah[2,3]

1 Laboratory of Molecular Biotechnology of Eukaryotes, Center of Biotechnology of Sfax, University of Sfax, Sfax, Tunisia, 2 Laboratory of Microbiology, Research Laboratory for Microorganisms and Human Disease LR03SP03, Habib Bourguiba University-Hospital, Sfax, Tunisia, 3 Faculty of Medicine, University of Sfax, Sfax, Tunisia, 4 Charles Nicolle Hospital, Laboratory of Microbiology, Tunis, Tunisia, 5 University of Tunis El Manar, Faculty of Medicine of Tunis, Tunis, Tunisia, 6 National Center Chalbi belkahia of Pharmacovigilance of Tunis, Laboratory of Clinical Pharmacology, Tunis, Tunisia, 7 University of Tunis El Manar, Faculty of Medicine of Tunis, Tunis, Tunisia

* wajdi.ayadi.cbs@gmail.com

## Abstract

Monitoring the emergence and propagation of SARS-COV2 variants, especially Omicron variants, remains a major concern in developing countries, including Tunisia. We here report lessons of simple approaches used to track prevalent Omicron variants in the city and district of Sfax, Tunisia, between June 2022 and April 2023. Initially, the screening approach was designed by selecting and verifying key SARS-CoV-2 mutations using publicly available sequencing data. Then, the analytical performance of the screening tests was rigorously assessed before being implemented on 227 confirmed COVID-19 cases. In a first stage, from June to September 2022, allele-specific (AS)-qPCR detection of deletions ΔHV69-70 (S gene) and ΔKSF141-143 (ORF1a gene) allowed identification of BA.5 as the predominant variant (128 out of 165 cases; 77.5%) which quickly replaced the pre-existing lineages BA.4 (15.7%) and BA.2 (6.7%). In a second stage, from October 2022 to April 2023, circulation of additional variants was demonstrated using concomitant detection of new relevant mutations by PCR-RFLP (n=62). Detection of mutations Y264H (ORF1b) and V445P/ G446S (S gene) resulted in the identification of 38 cases of the BQ.1 variant and 14 cases of the XBB variant, respectively. Further restriction analysis of the S gene was conducted to screen more recent sublineages, including CH.1.1. For all sequenced cases (n=115), our rapid detection assays showed perfect concordance with sequencing results in identifying SARS-CoV-2 variants. These findings highlight the potential of simple, cheap and proven methods for rapid genotyping and monitoring

**Data availability statement:** All relevant data are within the paper and its Supporting Information files.

**Funding:** The author(s) received no specific funding for this work.

**Competing interests:** The authors have declared that no competing interests exist.

of SARS-COV2 variants. Therefore, these methods appear as valuable tools for effective infection control and prevention in developing countries.

## Introduction

Throughout the COVID-19 pandemic, severe acute respiratory syndrome coronavirus 2 (SARS-CoV-2) has continuously evolved, leading to the emergence of multiple variants with an increased ability to circulate at high levels around the world [1].

The emergence of the Omicron variant (B.1.1.529) marked the onset of a new phase in the pandemic, which is characterized by the circulation of diverse sublineages and recombinants. Initially, three main lineages BA.1, BA.2 and BA.3 emerged independently in October 2021, followed by BA.4 in mid-December 2021 and BA.5 in early January 2022 [2]. Both BA.4 and BA.5 have identical Spike (S) gene sequences, differing from BA.2 by three additional mutations (ΔHV69-70, L452R, and F486V) and a reversion at position Q493. BA.4 also has unique mutations compared to BA.5, including P151S in the nucleocapsid (N) gene, L11F in the ORF7b gene, and a three-residue deletion (ΔKSF141-143) in the ORF1a gene [3].

Since the end of 2022, the epidemiological situation has been characterized by a very diverse landscape of multiple emerging sublineages and recombinant forms, particularly within BA.2 (such as BA.2.75, CH.1.1, and XBB) and BA.5 (including BQ.1 and BQ.1.1) [4,5]. Among them, BQ.1 and BQ.1.1 and XBB have demonstrated a high level of circulation worldwide due to their ability to evade monoclonal antibodies and immune protection established by previous natural infection and/or vaccination [6–8]. Both BQ.1 and BQ.1.1 harbor additional mutations beyond those found in BA.5, including K444T and N460K (AAA codon) in the receptor-binding domain (RBD) of the S gene, as well as Y264H in the ORF1b gene [7,9]. On the other hand, XBB encompasses a mixed genetic material of both parental BA.2 sublineages BJ.1 (BA.2.10.1.1) and BM.1.1.1 (BA.2.75.3.1.1.1), leading to increased complexity in the S gene. This recombinant form harbors distinct mutations such as V445P, G446S, and N460K (AAG codon) in the RBD domain [10]. XBB has given rise to several sublineages, including XBB.1.5 which is characterized by a growth advantage compared to previous variants [6,11].

Whole genome sequencing (WGS) using next-generation sequencing (NGS) has enabled the identification of emerging variants and continuous monitoring of the COVID-19 pandemic's evolution [12–15]. Despite considerable efforts to enhance sequencing capabilities, there are striking differences in the intensity of SARS-CoV-2 genomic surveillance worldwide. In the initial two years of the pandemic, nearly 80% of high-income countries managed to sequence at least 0.5% of their positive COVID-19 cases, whereas fewer than 50% of middle- and low-income countries reached this level of sequencing coverage [16,17]. Alternatively, several methods based on mutation-specific PCR have been successfully used to screen for SARS-CoV-2 variants, particularly those with high levels of circulation worldwide [18–23].

This study aimed to evaluate the effectiveness of a screening approach using allele-specific (AS)-qPCR and PCR-RFLP assays targeting Omicron's characteristic mutations to rapidly detect the main variants of SARS-CoV-2 circulating in Tunisia between June 2022 and April 2023.

## Materials and methods

### Samples

We included a total of 227 SARS-CoV-2 positive samples initially collected from June 2022 to April 2023 at the microbiology laboratory of Habib Bourguiba University Hospital, Sfax, Tunisia. The samples were obtained from a population with a nearly balanced gender distribution (56% female, 44% male), covering all age groups from newborns to 92 years (mean: 47 ± 22 years). The samples were initially extracted from nasopharyngeal swabs with AlphaPrep™ Viral DNA/RNA kit (Alphagene Co., Ltd, Korea) on the NC15 Plus instrument and amplified by real-time RT-PCR using either Wondfo 2019-nCoV Real-Time RT-PCR Assay (Guangzhou Wondfo Biotech Co., Ltd.) or ANDiS FAST SARS-CoV-2 RT-qPCR Detection Kit (3D Biomedicine Science & Technology Co., Ltd). These samples had ORF1ab and N quantification cycle (Cq) values comprised between 13 and 30. Ethical approval for the study was obtained by the local Ethics Committee of the Faculty of Medicine of Sfax (approval no. 06/24, date: April 19th 2024). The samples were fully anonymized before we accessed them from April 20, 2024 for research purposes and the Ethics committee waived the requirement for informed consent, as no details to identify individual participants were available after data collection.

### Screening design

To rapidly screen for circulating Omicron sublineages and recombinants, we designed an adaptable algorithm that can be modified based on the epidemiological landscape. We selected SARS-CoV-2 mutations based on their prevalence across different Omicron sublineages and the feasibility of targeting them within our approach.

From June to September 2022, two recurrent deletions were assessed using an allele-specific (AS)-qPCR: ΔHV69-70 (S gene) and ΔKSF141-143 (ORF1a gene). The ΔHV69-70 deletion distinguished BA.4 and BA.5 lineages from the BA.2 variant, whereas the ΔKSF141-143 deletion was used to differentiate between BA.4 and BA.5.

Between October 2022 and April 2023, the ΔHV69-70 deletion was targeted using AS-qPCR to differentiate between BA.5 and BA.2. Simultaneously, a new set of mutations was introduced and detected through PCR-restriction fragment length polymorphism (RFLP). This method was employed to detect the Y264H (ORF1b gene) and V445P/G446S (S gene) which identified Omicron BQ.1 and XBB variants, respectively. For cases where the variant remained undefined, additional analyses were conducted to examine the K444X (T, R, N, or M), L452Q, and N460K (AAG codon) in the S gene to identify CH.1.1 and BA.2.75 lineages (Fig 1).

Prior executing the proposed experimental design, the ability of the selected mutations to distinguish circulating SARS-CoV-2 sublineages was assessed using international sequence data published over two periods: the first period was from 1 June 2022 until September 2022 and the second period was between 1 October 2022 and 30 April 2023. The Cov-Spectrum platform was employed to extract the counts of sequences harboring deletions or wild-type non-deleted regions, as well as those containing or lacking restriction sites, for every lineage, based on aggregated data from the GISAID [24,25]. Subsequently, the sensitivity and specificity were calculated for each variant based on the selected genetic markers (deletions/wild-type non-deleted regions and restriction sites), according to the following equations:

$$\text{Sensitivity} = \left[\text{True Positive}/\left(\text{True Positive} + \text{False Negative}\right)\right] \times 100\%$$

$$\text{Specificity} = \left[\text{True Negative}/\left(\text{True Negative} + \text{False Positive}\right)\right] \times 100\%$$

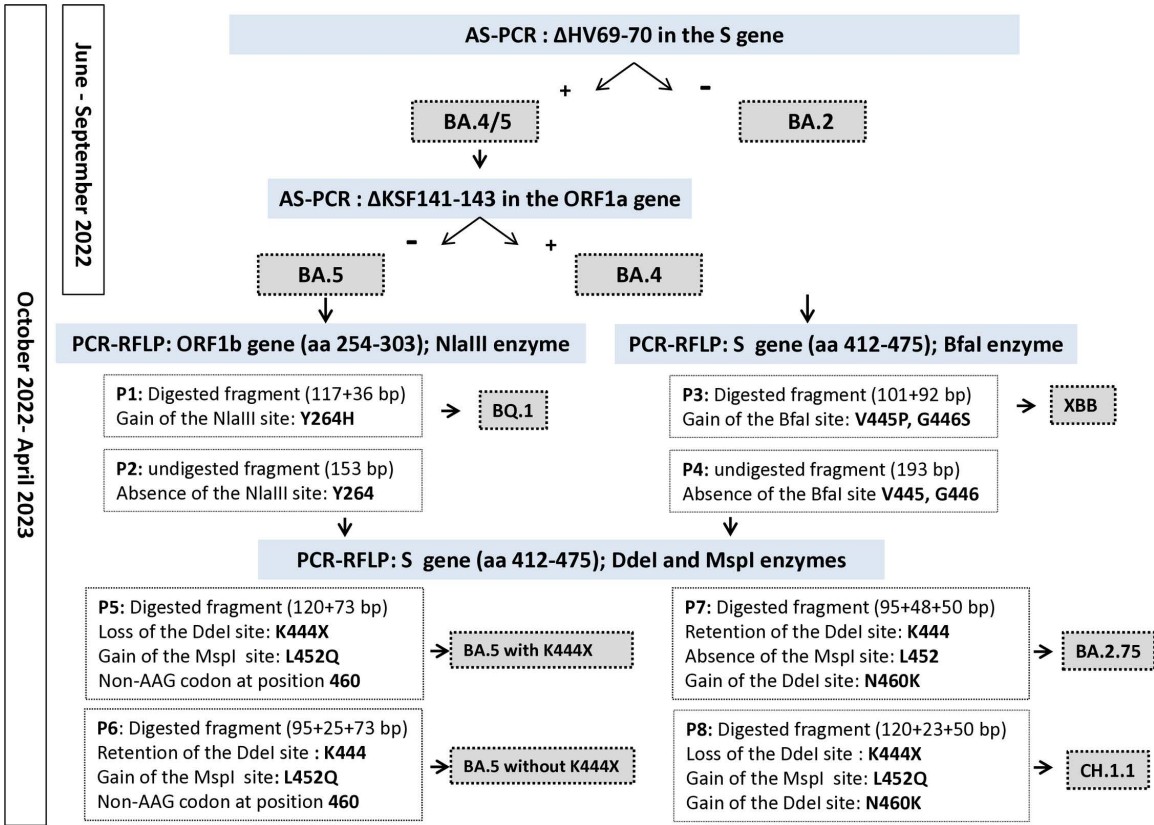

**Fig 1. Algorithm executed to detect the predominant circulating Omicron variants using consecutive AS-qPCR and PCR-RFLP assays targeting relevant mutations.** P1-8: different restriction profiles.

## Construction of recombinant plasmids

To evaluate the analytical sensitivity, specificity and reproducibility of our screening assays, six recombinant plasmids were constructed containing the main target sequences of this study: the KSF141-143 and ΔKSF141-143 in the ORF1a gene, the Y264H in the ORF1b gene and the HV69-70, ΔHV69-70 and V445P/G446S in the S gene. A fragment of each target sequence was separately amplified from previously sequenced samples, under the same conditions outlined in the PCR-RFLP sub-section of the Materials and methods. The PCR products were cloned in a pGEM-T vector (Promega), and then transformed into *E. coli*. Individual positive colonies were picked for growing in liquid medium, followed by plasmid DNA extraction using FavorPrep™ Plasmid DNA Extraction Mini Kit (Favorgen Biotech Corporation). The concentration of each recombinant plasmid was determined by Nanodrop (Spectrophotometer 2000) and the corresponding copy number (copies/μL) was calculated using the following equation:

$$\text{Plasmid copy number} = \left(\text{DNA concentration of plasmid (ng/}\mu\text{L)} \times 10^{-9} \times 6.022 \times 10^{23}\right) /$$
$$(\text{plasmid length (bp)} \times 660 \,(\text{g/mole of bp})).$$

Finally, serial ten-fold dilutions of each recombinant plasmid with a known copy number was generated and used as standards for the corresponding AS-qPCR or PCR-RFLP screening assay.

## Reverse transcription

First-strand cDNA synthesis was prepared from 4 µl of RNA extract in a final volume of 20 µl containing 1× reaction buffer, 0.2 pmol of each of the specific Reverse primers (Table 1), 20 nmol of each dNTP, 0.2 µmol of DTT and 100 U of SCRIPT Reverse Transcriptase (Jena Bioscience). The reaction was incubated at 50 °C for 30 min, followed by 70°C for 10 min for enzyme inactivation.

## AS-qPCR assays

For each AS-qPCR assay targeting the ΔHV69-70 or ΔKSF141-143 deletion, two different Forward primers at their 3' end (WT-F and 1Δ-F) and a common Reverse primer were designed to detect wild-type from mutant strains separately (Table 1). The qPCR reaction mix includes 1 µl of cDNA or plasmid DNA as a template, 2 pmol of each primer pair (Table 1), and 10 µl of 2 × TB Green Premix Ex Taq (Takara). Amplification was conducted on the Biorad-CFX96 using the following conditions: 95°C for 40 s, then 40 cycles of 94°C for 10 s and 60°C for 30 s. The specificity of each PCR product was confirmed by melting curve analysis from 65°C to 95°C, with a heating rate of 0.5°C/s. The analytical sensitivity of each AS-qPCR assay was evaluated using ten-fold serial dilutions of recombinant plasmid DNA, ranging from $5 \times 10^7$ to 5 copies/reaction. To assess the analytical specificity of the AS-qPCR assays, each primer pair was tested against the opposite genotype using ten-fold serial dilutions of recombinant plasmid DNA. Additionally, intra-assay and inter-assay variability of Cq values were assessed using the coefficient of variation (CV) in triplicate repeated tests.

## PCR-RFLP

For restriction analysis, two PCR reactions were conducted: one targeting codons 412–475 within the S gene and the other targeting codons 254–303 within the ORF1b gene (Table 1). All PCR reactions were performed on 1 µl of cDNA or plasmid DNA in a final reaction volume of 25 µl containing 1x PCR buffer, 0.15 µM of each primer, 200 µM of each dNTP, and 1.25 units of Dream Taq DNA polymerase (ThermoFisher Scientific). DNA amplification was carried out on the Bio-Gener thermal cycler with an initial denaturation at 94 °C for 2 min, 40 cycles of 30 s at 94 °C, 30 s at 60 °C, and 30 s at 72 °C, followed by a final extension of 5 min at 72 °C. All PCR products were then separated on a 2% agarose gel and visualized under UV light after staining with ethidium bromide.

Once successful amplification was confirmed, each PCR product specific to the ORF1b or S gene was subjected to enzymatic digestion using the appropriate restriction endonuclease, NlaIII or BfaI, respectively. The NlaIII enzyme recognizes the CATG restriction site created by the Y264H substitution in the ORF1b gene (TAT→CAT). Likewise, the BfaI

**Table 1. Summary of primer sequences used for detection of Omicron variants by AS-PCR, PCR-RFLP and PCR sequencing.**

| Assay | Target sequence (gene) | Primer sequence (5'-3') | Amplicon size |
|---|---|---|---|
| **AS-qPCR** | ΔHV69-70 (S) | S69-70-F: CTTGGTTCCATGCTATACATG. <br> SΔ69-70-F: CTTGGTTCCATGCTATCTC <br> S69-70-R: ATGGTAGGACAGGGTTATC. | 69 bp <br> 63 bp |
| | ΔKSF141-143 (ORF1a) | a141-143-F: GCGCCGATCTAAAGTCATTT <br> aΔ141-143-F: GGCGCCGATCTAGACTT <br> a141-143-R: CGTTAAGCTCACGCATGAG | 129 bp <br> 121 bp |
| **PCR-RFLP** | aa 412–475 (S) | S412-F: CAGGGCAAACTGGAAAT <br> S475-R: CGGCCTGATAGATTTCAGTTG | 196 bp |
| | aa 254–303 (ORF1b) | b254-F: CAAAGCCTTACATTAAGTGGG <br> b303-R: AGTTTGCACAATGCAGAATG | 153 bp |
| **PCR-sequencing** | aa 332–527 (S) | S332-F: ATTACAAACTTGTGCCCTTTT <br> S527-R: AGGTCCACAAACAGTTGCT | 588 bp |

enzyme targets the CTAG site generated by the V445P/G446S mutations (GTT→CCT and GGT→AGT, respectively). The analytical sensitivity of each reaction was evaluated using ten-fold serial dilutions of recombinant plasmid DNA, ranging from $5 \times 10^9$ to $5 \times 10^2$ copies/reaction in triplicate repeated tests.

Enzymatic reactions were carried out in a 20 µl reaction mixture containing 10 µl to 15 µl of PCR product, 1× appropriate buffer, and 10 units of each enzyme (ThermoFisher Scientific). After incubation at 37°C for 4 hours, the DNA fragments were resolved on 8% polyacrylamide gel and stained with ethidium bromide. The same protocol was used to identify the DdeI and MspI restriction sites in the amplified S gene fragment. The L452R (CTG→CGG) and N460K (AAT→AAG) substitutions introduce new MspI and DdeI restriction sites, respectively, while the K444X (T, R, N, or M) mutation results in the loss of the DdeI target sequence. The predicted restriction profiles and their associated Omicron variants are shown in Fig 1.

### NGS and Sanger sequencing

Out of the 227 samples included in the present study, 115 were subjected to partial (n=25) or whole-genome sequencing (n=90) as part of the national surveillance program for SARS-CoV-2 variants, which was implemented by the Ministry of Health in Tunisia. For NGS, libraries were prepared using the Illumina COVIDSeq test kit (Illumina Inc, USA). Briefly, the SARS-CoV-2 cDNA was amplified using the two sets of primers (COVIDSeq Primer Pool-1 & 2) then PCR amplicons subjected to the tagmentation procedure using the Enrichment Bead-Linked Transposomes (Enrichment BLT). Further amplification of ligated amplicons with the distinct pre-paired 10 base pair Index 1 (i7) and Index 2 (i5) adapters (IDT for Illumina-PCR) was performed. Finally, the pooled libraries were quantified using a Qubit 4.0 fluorometer (Invitrogen, Inc.) and were diluted to a final loading concentration of 100 pM, according to the iSeq 100 System. The 90 genome sequences were deposited in the Global Initiative on Sharing All Influenza Data (GISAID) with references shown in S1 Table. For Sanger sequencing, a partial region in the S gene encoding a sequence from codon 332 to codon 527 was amplified using S332-F/S527-R primer pairs (Table 1) and sequenced as previously described [26]. Briefly, PCR products were purified with HT Exo SAP-IT (ThermoFisher Scientific), sequenced using BigDye™ Terminator V.3.1 Cycle Sequencing kit (Applied Biosystems), purified with Invitrogen™ Dynabeads ™ Magnetic Beads (ThermoFisher Scientific) and analyzed on ABI 3500 Genetic Analyzer (Applied Biosystems). The corresponding Omicron variants were defined based on their key specific mutations, as shown in Table 2.

## Results

### Validation of the selected genetic markers

Using publicly available sequence data worldwide, we evaluated the ability of the genetic markers targeted by our screening approach to accurately identify the circulating Omicron variants in the two time periodes studied. Between June and September 2022, the ΔHV69-70 deletion had high sensitivity and specificity (>96%) in distinguishing BA.4/BA.5 lineages from BA.2. The combinations "ΔHV69-70 + ΔKSF141-143" and "ΔHV69-70 + wild-type KSF141-143" were shown to discriminate BA.4 and BA.5, respectively, with high sensitivity and specificity (>97%). From October 2022 to April 2023, the ΔHV69-70 deletion demonstrated elevated performance in screening BA.5/BQ.1 from BA.2/XBB (>95%). Furthermore, it was highlighted that the chosen genetic markers resulting in loss or gain of restriction sites had good sensitivity and specificity for screening BA.5 (non-BQ.1), BQ.1, XBB and CH. 1.1, respectively (Fig 2 and S2 Table).

### Analytical performance of screening assays

The limit of detection (LOD) of AS-qPCR was 5 copies/reaction for the ΔHV69-70 deletion and its corresponding wild-type sequence in the S gene, and 50 copies/reaction for both deleted and wild-type target sequences in the ORF1a gene (Fig 3). The coefficient of variation (CV) of Cq values ranged from 0.05 to 2.25 for intra-assay variability and from 0.30 to 3.63 for inter-assay variability (S3 Table). Regarding the analytical specificity of AS-qPCR, all primer pairs demonstrated

**Table 2. Key specific mutations examined through partial sequencing of S gene for identification of Omicron sublineages.**

| Positionᵃ | BA.2 | BA.4/BA.5 | BQ.1 | BQ1.1 | BA.2.75 | CH1.1 | XBB |
|---|---|---|---|---|---|---|---|
| G339 | D | D | D | D | H | H | H |
| R346 | | | | T | | T | T |
| L368 | | | | | | | I |
| K444 | | | T | T | | T | |
| V445 | | | | | | | P |
| G446 | | | | | S | S | S |
| L452 | | R | R | R | | R | |
| N460ᵇ | | | K (AAA) | K (AAA) | K (AAG) | K (AAG) | K (AAG) |
| F486 | | V | V | V | | S | P |
| F490 | | | | | | | S |
| Q493 | R | | | | | | |

ᵃAccording to the reference sequence SARS-CoV-2/Wuhan-Hu-1 (NC_045512)

ᵇThe N460 mutation lists codon nucleotides in parentheses

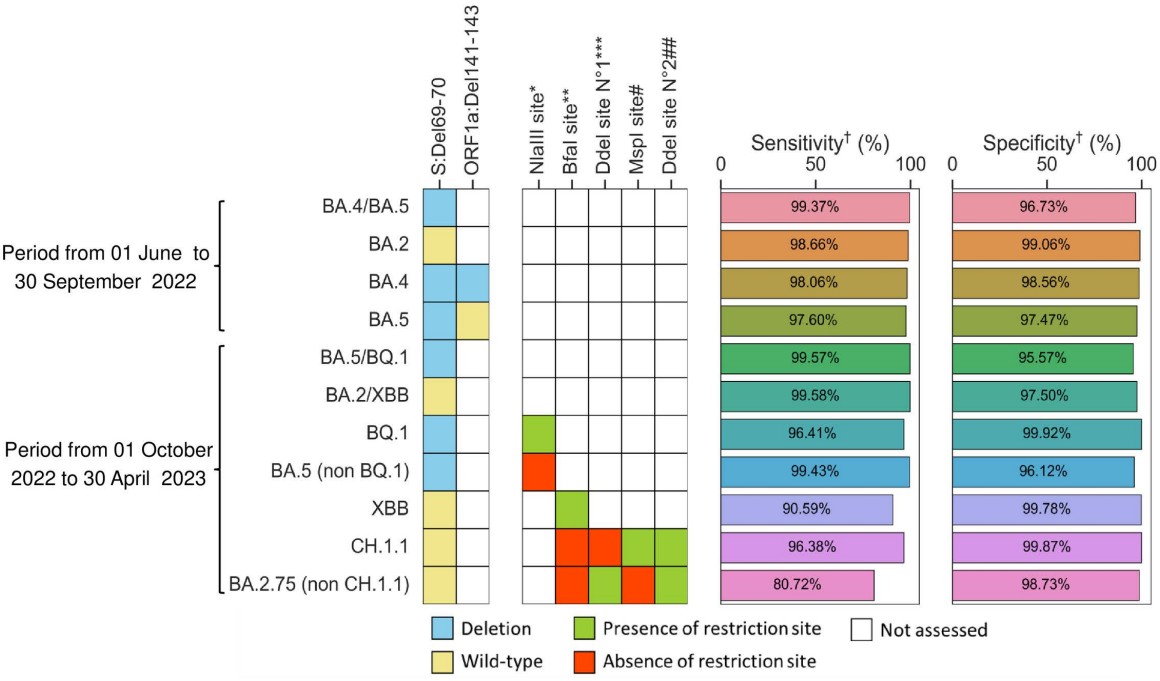

**Fig 2. Sensitivity and specificity of selected genetic markers based on sequence data from GISAID.** *NlaIII site: "CATG" from nucleotide 14257 to 14260 (Y264H in ORF1b). **BfaI site: "CTAG" from nucleotide 22896 to 22899 (V445P-G446S in Spike). ***DdeI site N°1: "CTNAG" from nucleotide 22890 to 22894 (K444 in Spike). #MspI site: "TTGG" from nucleotide T22915C to 22917 (L452Q in Spike). ##DdeI site N°2: "CTNAG" from nucleotide 22890 to 22894 (N460K in Spike). †The data used to calculate the sensitivity and specificity for each lineage are provided in S2 Table.

perfect discrimination between mutant and wild-type sequences, with no detectable cross-reactivity using diluted recombinant plasmid DNA below $5.10^6$ copies/reaction. Nevertheless, cross-reactivity was observed with Cq values > 32.9 at higher concentrations ($5 \times 10^6$ and $5 \times 10^7$ copies/reaction) for primer pairs targeting the wild-type and deleted HV69-70

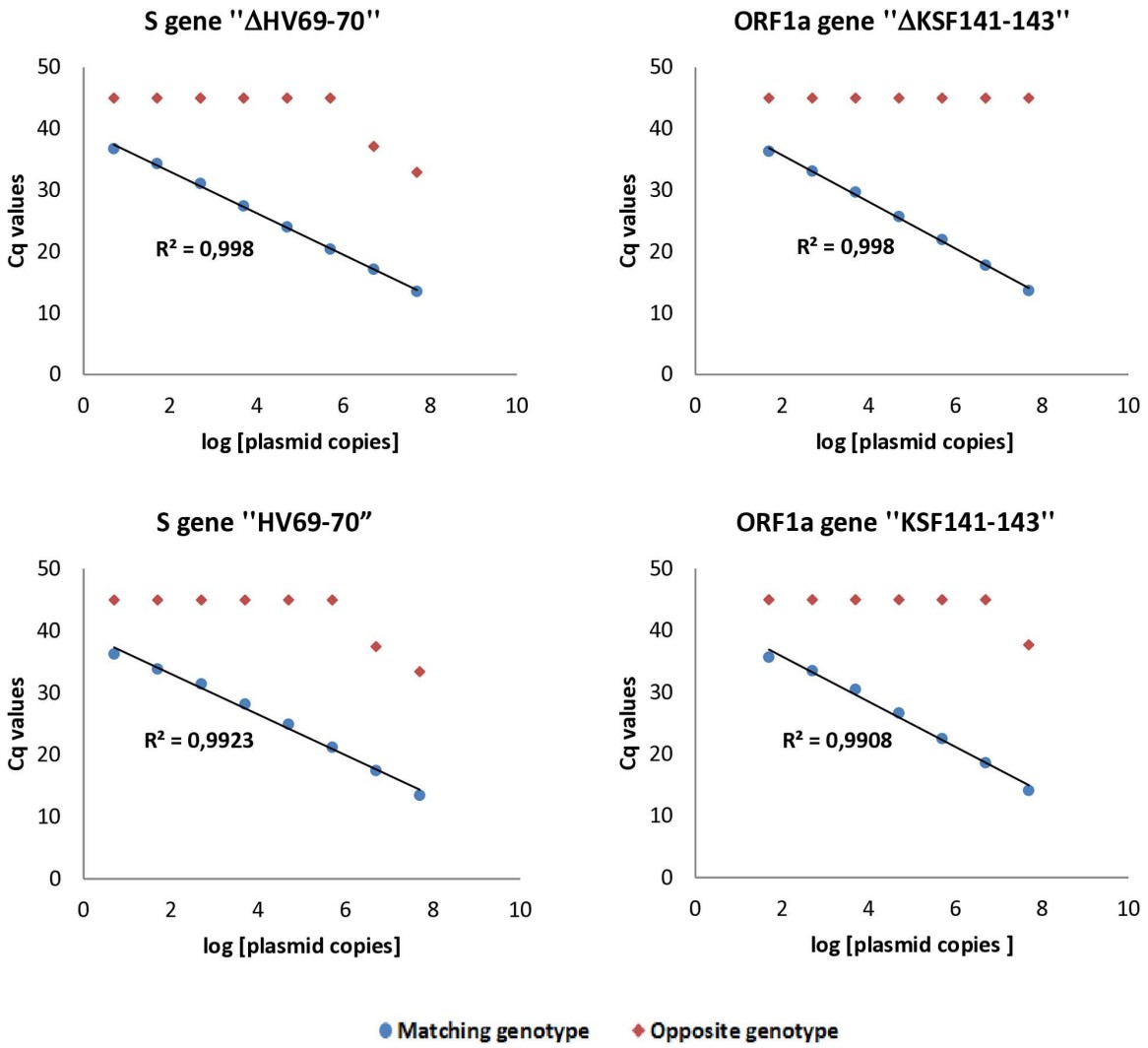

**Fig 3. Analytical sensitivity and specificity of AS-qPCR assays targeting the deleted forms and their corresponding wild-type sequences.** Cq values based on 10-fold serial dilutions of recombinant plasmid DNA ranging from $5x10^7$ to 5 copies per reaction. Cq values for plasmids without amplification are set arbitrarily to 45. The plasmid copies are presented as logarithm to base 10 (log). $R^2$: coefficient of determination.

regions in the S gene, and at $5 \times 10^7$ copies/reaction for the primer pair targeting the wild-type KSF141-143 in the ORF1a gene. To address this, clinical samples with Cq values < 20 in routine RT-PCR diagnosis were subjected to a $10^{-2}$ cDNA dilution before analysis with the developed AS-qPCR assays.

For PCR-RFLP assays, the LOD was $5 \times 10^4$ copies/reaction for the Y264H mutation in the ORF1b gene and the V445P/G446S mutations in the S gene (S1 Fig). The repeatability of the PCR-RFLP assays was confirmed, as successful amplification was achieved in all triplicate repeated tests.

## Omicron BA.4 and BA.5 screening

During the period from June to September 2022, the first AS-qPCR was applied to rapidly screen SARS-CoV-2 variants harboring the ΔHV69-70 deletion from variants lacking it; mainly Omicron BA.4/BA.5 from BA.2. This deletion was

identified in 154 out of 165 cases that were subsequently classified as Omicron BA.4/BA.5 according to the epidemiological status during the summer of 2022. On the other hand, the corresponding HV69-70 wild-type sequence was detected in 11 samples which were subsequently classified as Omicron BA.2 variant. Among the 154 strains harboring the ΔHV69-70 deletion, the ΔKSF141-143 deletion was identified in 26 cases, which were labeled as BA.4. The remaining 128 cases were shown to have the corresponding wild-type KSF141-143 sequence and were subsequently classified as BA.5.

## Screening of BA.2 and BA.5 descendant sublineages

From October 2022 to April 2023 (n=62), AS-qPCR testing the ΔHV69-70 deletion and the ΔKSF141-143 deletion defined 44 cases as BA.5 and 18 cases as BA.2. However, BA.4 sublineage was not detected during this second period. To distinguish BQ.1 from other BA.5 sublineages, the PCR-RFLP targeting the Y264H mutation in the ORF1b gene was successfully performed on 41 out of 44 cases. The Cq values of the three failed samples were above 28 in routine diagnostic RT-PCR. BQ.1 sublineages were found in almost all cases (n=38), while the remaining three cases were labeled as other BA.5 sublineages (Fig 4A).

Concerning the BA.2 sublineages and the recombinants, PCR-RFLP analysis based on the two successive mutations V445P and G446S of the S gene revealed the recombinant XBB variants in 14 out of 18 cases (Fig 4B). For undefined cases of the BA.2 and BA.5 descendant sublineages, additional restriction analysis of the amplified S fragment was performed to detect the following mutations: K444X (T, R, N or M), L452R and N460K (AAG). Among the non-XBB cases of BA.2 sublineages (n=4), the CH.1.1 sublineage which carries all three mutations was detected in two cases and the parental BA.2.75 sublineage which only harbors the N460K (AAG) substitution was identified in the remaining two cases. Concerning the non-BQ.1 cases within the BA.5 sublineages, all three cases showed the same restriction profile, indicating that they carried the K444X and L452R substitutions (Fig 4C).

## Concordance between screening and sequencing results

The SARS-CoV-2 sequences obtained for 115 out of 227 samples were used to evaluate the ability of our described screening tests to correctly detect SARS-CoV-2 variants. The comparative analysis between screening tests and

**Fig 4. Representative results of PCR-RFLP analysis showing specific mutations of Omicron sublineages. A**: PCR amplification and NlaIII digestion of OFR1b gene (aa 254-303) detecting the Y264H mutation characterizing BQ.1 sublineages in lanes 1, 2, 4 and 5. **B**: PCR amplification and BfaI digestion of S gene (aa 412-475) detecting the two successive mutations V445P and G446S characterizing XBB sublineages in lanes 6, 7, 8, 9, 12 and 13. **C**: PCR amplification and MspI/DdeI digestion of S gene (aa 412-475) detecting the two mutations K444X (T, R, N or M) and L452Q of the BA.5 descendent sublineages in lanes 14, 15 and 16; the N460K (AAG) mutation characterizing the parental BA.2.75 sublineage in lanes 17 and 18; the three mutations K444X (T, R, N or **M**), L452Q and N460K (AAG) characterizing CH1.1. sublineage in lane 19. Lane L1: 50 bp DNA ladder. Lane L2: 100 bp DNA ladder.

sequencing showed perfect concordance to attribute the three lineages Omicron BA.2 (n=5), BA.4 (n=7) and BA.5 (n=48) collected from June to September 2022 as well as BQ.1 (n=35), XBB (n=14), CH.1.1 (n=2) and BN.1.3 (sublineage of parental BA.2.75, n=2) collected from October 2022 to April 2023. Of note, two out of three BA.5 (non-BQ.1) collected during the second period were partially sequenced in the S gene, revealing the presence of K444M and L452R substitutions, consistent with their corresponding restriction profile (K444X and L452R).

Additionally, the sequencing results provided more details on the designation of Omicron sublineages, including BQ1.1 (n=16), XBB.1.9 (n=8), XBB.1.5 (n=2) and XBB.1.11 (n=1).

## Discussion

Following the emergence of each new SARS-CoV-2 variant with a high potential of infectivity and transmissibility requiring close monitoring, several genotyping methods based on its relevant mutations have been designed and validated as alternative tools to sequencing.

In this context, our previous study demonstrated the capacity of a simple and flexible screening method to monitor the introduction and local spread of the first two Omicron variants BA.1 and BA.2 in Tunisia, since January 2022 [26]. Given that Omicron lineages have evolved into numerous branching sublineages, this study aimed to develop molecular tests for the rapid screening of subsequent Omicron variants. This includes BA.4, BA.5, and its descendant sublineages BQ.1/BQ.1.1, as well as XBB, a recombinant form arising from two BA.2 sublineages.

To address this need, we first developed and analytically validated AS-qPCR assays to detect the ΔHV69-70 and ΔKSF141-143 deletions, along with their wild-type sequences, in the S and ORF1a genes, respectively. These assays demonstrated analytical performance comparable to previously reported methods for detecting SARS-CoV-2 variants [19,27,28], highlighting their potential value for screening Omicron lineages and descendant sublineages.

The ΔHV69-70 deletion in the S gene, initially associated with the Alpha variant, has also been effective in differentiating between Omicron sublineages, such as BA.1 from BA.2 [29–31] and BA.4/BA.5 from BA.2 [32]. Here, our screening method, which utilizes the ΔHV69-70 deletion as a primary marker, demonstrated a predominance of BA.4/BA.5 over BA.2 from June to September 2022 (90.3% vs. 9.7%). To differentiate between the closely related BA.4 and BA.5 sublineages, we selected the ΔKSF141-143 deletion in the ORF1a gene. This deletion has been previously employed to identify BA.4 among Omicron sublineages in Taiwanese patients from April to December 2022 [33] and in Vietnamese patients from January to September 2022 [34]. Our findings showed a lower prevalence of BA.4 compared to BA.5 (17% vs. 83%), which is in line with previous studies [33,34]. The difference in the spread potential of these two sublineages is unrelated to the S glycoprotein, as they share identical S gene sequences. Further research into the genetic variations between BA.4 and BA.5, especially from the M gene to the 3′ end of the genome [3], may provide more insights into this point.

In late 2022, the epidemiological landscape saw notable shifts with the emergence of new Omicron sublineages, which demonstrated increased immune evasion and decreased effectiveness of monoclonal antibody treatments [6–8]. Holland et al. developed a digital PCR test panel targeting specific mutations in the RBD of the S gene: K444T, N460K (AAA) and F486V for detecting BQ.1, and R346T, N460K (AAG) and F486S for detecting XBB [35]. In this study, we designed and validated a simple and cost-effective PCR-RFLP method for rapid screening of circulating Omicron subvariants with minimal equipment. Although the restriction method is less sensitive than qPCR-based approaches and may be limited by the creation or loss of restriction sites, it has been previously employed to differentiate SARS-CoV-1 from SARS-CoV-2 [36] and to distinguish SARS-CoV-2 variants based on key substitutions in the S gene, such as K417N, N440K, and E484K [37,38]. Our PCR-RFLP assay, which targets the Y264H substitution (ORF1b gene), successfully identified the BQ.1 sublineage in nearly all samples collected from October 2022 to January 2023.

Since early February 2023, there has been a considerable rise in the prevalence of BA.2 sublineages and recombinant forms, with XBB being the predominant variant identified based on V445P and G446S mutations. Furthermore, our

PCR-RFLP assay, which targets the K444X (T, R, N, or M), L452R, and N460K (AAG codon) mutations in the S gene, effectively detected the less frequent SARS-CoV-2 sublineages CH.1.1 (n=2) and BA.2.75 (n=2).

Partial S gene (codons 339–520) and whole genome sequencing findings perfectly aligned with the results of our screening approach, confirming its accuracy. Additionally, the effectiveness of the selected genetic markers in this approach was validated by evaluating their performance using global SARS-CoV-2 sequence data from the GISAID database. Our method offers a simple and cost-effective solution for high-throughput screening of SARS-CoV-2 positive samples, improving the capacities to track newly emerged variants, particularly in developing countries. It can be easily implemented in most laboratories with moderate technical expertise and minimal equipment investment. Additionally, this PCR-based approach can be adapted to detect new Omicron variants, such as EG.5 [39] and JN.1 [40], which carry the F456L (gain of the MseI site) and N450D (gain of the PsiI site) substitutions in the S gene, respectively.

Consistent with previous findings, Tunisia exhibited variant dynamics similar to those observed in neighboring North African countries and parts of Europe [26,41], reflecting regional transmission patterns shaped by travel and epidemiological factors. Furthermore, our screening approach successfully identified variants that were more prevalent in other regions, such as BA.2.75, which was widespread in parts of Asia [42]. This underscores the potential of our method for detecting variants beyond the Tunisian context.

In conclusion, our study highlights the effectiveness of a rapid, simple, cost-effective and adaptable screening approach for tracking Omicron variants. The implementation of such alternative screening tools is recommended to facilitate prompt monitoring of epidemiological trends and mitigate their impact on clinical care.

## Supporting information

**S1 Table. GISAID accession numbers of genome sequences.**
(XLSX)

**S2 Table. GISAID sequence count obtained through Cov-spectrum platform for sensitivity and specificity calculations.**
(XLSX)

**S3 Table. Coefficient of variation of the developed AS-qPCR assays.**
(DOCX)

**S1 Fig. Analytical sensitivity of PCR-RFLP assays using ten-fold serial dilutions of recombinant plasmid DNA ranging from $5 \times 10^9$ to $5 \times 10^2$ copies per reaction. A:** PCR amplification and NlaIII digestion of OFR1b gene (aa 254–303) detecting the Y264H mutation. **B**: PCR amplification and BfaI digestion of S gene (aa 412–475) detecting the two successive mutations V445P and G446S. L: 100 bp DNA ladder.
(TIF)

**S1 Raw image. Raw image files of polyacrylamide gels.**
(PDF)

## Acknowledgments

The authors acknowledge the technical staff of the Laboratory of Microbiology of the Habib Bourguiba University Hospital of Sfax for their efforts in COVID-19 diagnosis and the members of the Laboratory of Molecular Biotechnology of Eucaryotes, at the Center of Biotechnology of Sfax, for funds and support. We also acknowledge the staff of the Laboratory of Human Genetics, Hedi Chaker University Hospital of Sfax for their support in performing SARS-CoV-2 partial sequencing. We gratefully acknowledge all data contributors, i.e., the Authors and their Originating laboratories responsible for

obtaining the specimens, and their Submitting laboratories for generating the genetic sequence and metadata and sharing via the GISAID Initiative, on which this research is based.

## Author contributions

**Conceptualization:** Wajdi Ayadi, Lamia Feki-Berrajah.

**Data curation:** Fahmi Smaoui, Sana Ferjani, Zaineb Hamzaoui, Mouna Ben Sassi, Sameh Trabelsi.

**Formal analysis:** Fahmi Smaoui, Awatef Taktak.

**Investigation:** Wajdi Ayadi, Sana Ferjani, Azza Hadj Sassi, Mouna Ben Sassi, Sameh Trabelsi.

**Methodology:** Saba Gargouri, Azza Hadj Sassi.

**Software:** Fahmi Smaoui, Sana Ferjani, Zaineb Hamzaoui, Awatef Taktak.

**Supervision:** Saba Gargouri, Amel Chtourou, Houda Skouri-Gargouri, Ali Gargouri, Ilhem Boutiba-Ben Boubaker, Héla Karray-Hakim, Raja Mokdad-Gargouri, Lamia Feki-Berrajah.

**Validation:** Amel Chtourou, Houda Skouri-Gargouri, Ilhem Boutiba-Ben Boubaker, Lamia Feki-Berrajah.

**Writing – original draft:** Wajdi Ayadi, Fahmi Smaoui.

**Writing – review & editing:** Ali Gargouri, Ilhem Boutiba-Ben Boubaker, Héla Karray-Hakim, Raja Mokdad-Gargouri, Lamia Feki-Berrajah.

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
