## [Decision Letter · Decision Letter 0]

2 Jan 2025

PONE-D-24-46700Use of allele-specific qPCR and PCR-RFLP analysis for rapid detection of the SARS-CoV-2 variants in Tunisia: a cheap flexible approach adapted for developing countriesPLOS ONE

Dear Dr. Ayadi,

Thank you for submitting your manuscript to PLOS ONE. After careful consideration, we feel that it has merit but does not fully meet PLOS ONE’s publication criteria as it currently stands. Therefore, we invite you to submit a revised version of the manuscript that addresses the points raised during the review process.

We look forward to receiving your revised manuscript.

Kind regards,

Huseyin Tombuloglu

Academic Editor

PLOS ONE

Journal Requirements:

Reviewers' comments:

Reviewer's Responses to Questions

**Comments to the Author**

1. Is the manuscript technically sound, and do the data support the conclusions?

Reviewer #1: Yes

Reviewer #2: Yes

2. Has the statistical analysis been performed appropriately and rigorously? 

Reviewer #1: N/A

Reviewer #2: No

3. Have the authors made all data underlying the findings in their manuscript fully available?

Reviewer #1: Yes

Reviewer #2: Yes

4. Is the manuscript presented in an intelligible fashion and written in standard English?

Reviewer #1: Yes

Reviewer #2: Yes

5. Review Comments to the Author

Reviewer #1: ''Use of allele-specific qPCR and PCR-RFLP analysis for rapid detection of the SARS-CoV-2 variants in Tunisia: a cheap flexible approach adapted for developing countries'' has mentioned and explained the detection of allele and sublineage specific screening of SARS-CoV-2 Omicron variants. Restriction in sample collection was disadvantage of the study.

Additionally:

- Font type of Table 1 is different,

- In Figure 3 'October' has been written as 'Octobre' mistakenly.

- In lane 263, text looks selected and highlighted.

- Even the title of article involves name of Tunisia, possible ethnical or local factors relevant to Tunisia and comparison with others haven't been mentioned. Discussion part would be getting stronger depending on this detail.

Reviewer #2: The manuscript by Ayadi et al reports alternative methods for monitoring the SARS-CoV-2 variants. These methods seemed to be more suitable in the developing countries. I have several concerns:

1. When applying the as-PCR and PCR-RFLP for virus variant screening, it should be more cautious on determination the indies that could affect the results, including sample size, age and sex distribution of the enrolled samples etc. Although the authors declared that their variant monitoring results are in concordance with the epidemiological surveillance data, this is more like a concordance.

2. The authors should provide more details on the as-PCR and PCR-RFLP methods applied in the present study, how about their limit of detection, specificity, reproducibility etc.?

3. The structure of the manuscript is a little strange and recommended to be re-organized to make the manuscript more easy to follow.

6. PLOS authors have the option to publish the peer review history of their article (what does this mean? ). If published, this will include your full peer review and any attached files.

**Do you want your identity to be public for this peer review?** For information about this choice, including consent withdrawal, please see our Privacy Policy .

Reviewer #1: No

Reviewer #2: **Yes: ** Jianguo Li

---

## [Author Response · Author response to Decision Letter 0]

19 Feb 2025

Responses to Reviewer 1's Comments:

'Use of allele-specific qPCR and PCR-RFLP analysis for rapid detection of the SARS-CoV-2 variants in Tunisia: a cheap flexible approach adapted for developing countries'' has mentioned and explained the detection of allele and sublineage specific screening of SARS-CoV-2 Omicron variants. Restriction in sample collection was disadvantage of the study.

We greatly appreciate the reviewer’s insightful summary of our findings. The sample collection in our study encompasses nearly all SARS-CoV-2 positive samples detected at the microbiology laboratory of Habib Bourguiba University Hospital, Sfax, Tunisia, from June 2022 to April 2023. Samples with Cq values greater than 30 during routine RT-PCR diagnosis were excluded. Notably, we observed a decrease in the number of nasopharyngeal samples received during the study period compared to the earlier years of the pandemic.

Additionally:

- Font type of Table 1 is different

The font type in Table 1 has been corrected to match the rest of the manuscript for consistency.

-In Figure 3 'October' has been written as 'Octobre' mistakenly.

Thank you for pointing this out. The typographical error has been corrected from 'October' to 'October' in Figure 3. Following the requested reorganization of the manuscript, the figure number has been changed to Figure 2 in this new version.

-In lane 263, text looks selected and highlighted.

The unintentionally highlighted text in the previous version has been corrected in the revised manuscript.

-Even the title of article involves name of Tunisia, possible ethnical or local factors relevant to Tunisia and comparison with others haven't been mentioned. Discussion part would be getting stronger depending on this detail.

Thank you for this insightful suggestion. Based on our data and other previous findings, we conclude that the variant epidemiology in Tunisia has been largely influenced by the country's strong travel connections with Europe and other North African nations. We have expanded the following paragraph in the Discussion section (lines 406-411): “Consistent with previous findings, Tunisia exhibited variant dynamics similar to those observed in neighboring North African countries and parts of Europe [26,41], reflecting regional transmission patterns shaped by travel and epidemiological factors. Furthermore, our screening approach successfully identified variants that were more prevalent in other regions, such as BA.2.75, which was widespread in parts of Asia [42]. This underscores the potential of our method for detecting variants beyond the Tunisian context”.

Responses to Reviewer 2's Comments:

1/ When applying the as-PCR and PCR-RFLP for virus variant screening, it should be more cautious on determination the indies that could affect the results, including sample size, age and sex distribution of the enrolled samples etc. Although the authors declared that their variant monitoring results are in concordance with the epidemiological surveillance data, this is more like a concordance.

Thank you for raising this point. We have provided more details regarding these factors in lines 103-105. Overall, the samples were obtained from a population with a nearly balanced gender distribution (56% female, 44% male), covering all age groups from newborns to 92 years (mean: 47 ± 22 years). We believe that these demographic factors are unlikely to influence the results of variant screening, as current evidence indicates that SARS-CoV-2 lineage distribution is primarily shaped by viral evolution and epidemiological dynamics rather than host demographic characteristics.

2/ The authors should provide more details on the as-PCR and PCR-RFLP methods applied in the present study, how about their limit of detection, specificity, reproducibility etc?

We fully agree with the reviewer. The analytical sensitivity, specificity, and reproducibility of our screening tests have been demonstrated using recombinant plasmids containing the main target sequences of this study. We have included the necessary details in the Materials and Methods section (lines 148-165, 183-188 and 208-210), in results section (lines 269-291) and discussion section (lines 356-360).

3/ The structure of the manuscript is a little strange and recommended to be re-organized to make the manuscript more easy to follow.

The structure of the manuscript has been reorganized to enhance clarity and make it easier to follow. We adopted the following order: first, Validation of the selected genetic markers using publicly available global sequence data; second, Analytical performance of screening assays; followed by Omicron BA.4 and BA.5 screening, Screening of BA.2 and BA.5 descendant sublineages, and finally, Concordance between screening and sequencing results

The repositioned paragraphs are highlighted in blue within the text for your convenience.

---

## [Decision Letter · Decision Letter 1]

10 Mar 2025

Use of allele-specific qPCR and PCR-RFLP analysis for rapid detection of the SARS-CoV-2 variants in Tunisia: a cheap flexible approach adapted for developing countries

PONE-D-24-46700R1

Dear Dr. Ayadi,

We’re pleased to inform you that your manuscript has been judged scientifically suitable for publication and will be formally accepted for publication once it meets all outstanding technical requirements.

Kind regards,

Huseyin Tombuloglu

Academic Editor

PLOS ONE

Additional Editor Comments (optional):

Please double-check punctuation and some terminologies (SARS-CoV-2) and be sure they are written correctly.

Reviewers' comments:

Reviewer's Responses to Questions

**Comments to the Author**

1. If the authors have adequately addressed your comments raised in a previous round of review and you feel that this manuscript is now acceptable for publication, you may indicate that here to bypass the “Comments to the Author” section, enter your conflict of interest statement in the “Confidential to Editor” section, and submit your "Accept" recommendation.

Reviewer #2: All comments have been addressed

2. Is the manuscript technically sound, and do the data support the conclusions?

Reviewer #2: Partly

3. Has the statistical analysis been performed appropriately and rigorously? 

Reviewer #2: Yes

4. Have the authors made all data underlying the findings in their manuscript fully available?

Reviewer #2: Yes

5. Is the manuscript presented in an intelligible fashion and written in standard English?

Reviewer #2: Yes

6. Review Comments to the Author

Reviewer #2: All of my comments have been addressed, I have no more question. I recommend to accept the manuscript.

7. PLOS authors have the option to publish the peer review history of their article (what does this mean? ). If published, this will include your full peer review and any attached files.

**Do you want your identity to be public for this peer review?** For information about this choice, including consent withdrawal, please see our Privacy Policy .

Reviewer #2: **Yes: ** Jianguo Li

---

## [Editor Report · Acceptance letter]

PONE-D-24-46700R1

PLOS ONE

Dear Dr. Ayadi,

I'm pleased to inform you that your manuscript has been deemed suitable for publication in PLOS ONE. Congratulations! Your manuscript is now being handed over to our production team.

Kind regards,

on behalf of

Dr. Huseyin Tombuloglu

Academic Editor

PLOS ONE